# A Foundation Model Approach to Particle Accelerator Operational Data

**Mahindra Singh Rautela** [1]   **Alexander Scheinker** [1]

## Abstract

Large-scale scientific facilities, such as particle accelerators, are complex systems composed of thousands of interacting components that must operate collectively to support diagnostics and control and to ensure safe, reliable operation. These facilities rely on extensive sensor networks that generate large volumes of heterogeneous and noisy data, often sampled at different rates across subsystems. In this paper, we adopt the foundation-model paradigm to learn general-purpose sensor representations from historical accelerator data through self-supervised pretraining. We introduce SensOFormer, a denoising masked transformer with a Perceiver-style encoder–decoder architecture designed to handle variable sensor sets, accommodate variable sampling rates, and learn robust representations from noisy measurements. The model is pretrained on multiple particle-accelerator datasets and transferred to downstream tasks, including missing-data imputation, anomaly detection, cavity identification, and fault identification. Through extensive evaluations and comparisons, we demonstrate that a single self-supervised model can learn reusable representations for diverse operational tasks in large-scale scientific facilities.

## 1. Introduction

*Particle accelerator telemetry presents a foundation-model opportunity.* Particle accelerators are large-scale scientific facilities that enable a broad range of fundamental studies and scientific applications. Delivering these capabilities requires thousands of tightly coupled components to operate under strict stability, safety, and reliability constraints. Modern accelerators are therefore instrumented with extensive sensors and control systems that continuously monitor beam conditions, radio-frequency systems, magnets, vacuum, cooling, timing, and other subsystems (Chao et al., 2023). These instruments generate massive streams of operational telemetry that are noisy, heterogeneous, and distributed across the facility. As a result, core operational tasks such as missing-data reconstruction, anomaly detection, machine tuning, and control decision support depend critically on learning useful structure from high-dimensional sensor measurements (Edelen & Huang, 2024). The abundance of accelerator telemetry therefore motivates a foundation-model approach: learning general-purpose sensor representations from large-scale unlabeled operational data that can be adapted to diverse downstream accelerator tasks (Bommasani et al., 2021).

*Challenges with operational data.* Accelerator telemetry differs substantially from curated multivariate time-series benchmarks (Li et al., 2023). Operational data are inherently heterogeneous, collected from many subsystem-specific diagnostics (Tennant et al., 2020; St. John et al., 2021; Krymova et al., 2023; Liang et al., 2025; Radaideh et al., 2022). Each data acquisition (DAQ) segment may contain hundreds to thousands of channels, and at facility scale the number of available process variables can be much larger (St. John et al., 2021). Moreover, the available sensors, their ordering, and their grouping can vary across sectors, subsystems, and acquisitions. Consequently, the semantic identity of a channel is determined by sensor metadata, physical location, and subsystem rather than by its column index (Krymova et al., 2023). Second, accelerator measurements are asynchronous: sensors are sampled at different rates and logged over variable-length windows (Tennant et al., 2020). Third, accelerators are time-varying systems whose operating distributions drift with machine configuration, component aging, calibration changes, and seasonal or operational conditions (Scheinker et al., 2023). Finally, the measurements are noisy with noise statistics that are difficult to specify a priori (Liang et al., 2025).

*Model design requirements and SensOFormer.* To address these challenges, SensOFormer combines an architecture with a pretraining strategy tailored to heterogeneous accelerator telemetry. The model uses a latent transformer with a Perceiver-style set encoder and set decoder: the encoder is permutation-invariant across input channels, enabling variable sensor sets without fixed column semantics

[1]Instrumentation and Controls Group (AOT-IC), Los Alamos National Laboratory, United States. Correspondence to: Mahindra Rautela <mrautela@lanl.gov>.

*Proceedings of the 43rd International Conference on Machine Learning, Seoul, South Korea. PMLR 306, 2026. Copyright 2026 by the author(s).*

(Jaegle et al., 2021; Lee et al., 2019), while the decoder is permutation-equivariant, preserving the batch-specific identities and locations of queried outputs. This encoder–transformer–decoder design supports flexible channels and sequence lengths without data or activation padding. SensOFormer is pretrained with a masked denoising reconstruction objective. Structured masks and additive noise are used to corrupt the input, and the model learns to reconstruct the clean masked values from temporal and cross-channel context. This encourages the model to learn robust representations rather than sensor-specific noise patterns (Batson & Royer, 2019; Li et al., 2023). Details of the pretraining objective, architecture, and downstream tasks are provided in Appendix B.

**Contributions.** This work introduces SensOFormer, to the best of our knowledge, the first foundation model designed for particle-accelerator operational telemetry. We pretrain on a diverse multi-facility benchmark spanning LANSCE, LHC, SNS, LCLS, and CEBAF datasets, and demonstrate transfer across heterogeneous accelerator tasks, including zero-shot missing-data imputation, anomaly detection, cavity identification, and fault identification. We also characterize SensOFormer's scaling behavior with respect to model capacity and pretraining data volume.

## 2. Related Work

**Scientific ML in particle accelerators.** Machine learning has become an increasingly important tool for particle accelerators, with applications in beam-dynamics modeling, diagnostics, optimization, control, tuning, physics-constrained simulation, and uncertainty quantification (Rautela et al., 2024; Edelen & Huang, 2024; Scheinker & Williams, 2025; Scheinker, 2025; Saxena et al., 2025; Roussel et al., 2021; Edelen et al., 2020; Wan et al., 2025; Leon & Scheinker, 2024; Mishra et al., 2021; Garcia-Cardona & Scheinker, 2024). In operational settings, ML has enabled adaptive virtual diagnostics at Los Alamos Neutron Science Center (LANSCE at Los Alamos National Laboratory) (Scheinker, 2021; Scheinker & Williams, 2025), superconducting RF cavity-fault localization and classification at Continuous Electron Beam Accelerator Facility (CEBAF at Jefferson Laboratory) (Tennant et al., 2020), anomaly detection in high-voltage converter modulator signals at Spallation Neutron Source (SNS at Oak Ridge National Laboratory) (Radaideh et al., 2022), beam-loss prediction at the Large Hadron Collider (LHC at CERN) (Krymova et al., 2023), and RF station fault identification at Linac Coherent Light Source (LCLS at Stanford Linear Accelerator Laboratory) using beam-position-monitor and high-frequency RF phase data (Humble et al., 2022; Liang et al., 2025). While these efforts show the effectiveness of specialist ML models, accelerator facilities share recurring RF, magnet, vacuum, timing, diagnostic, and control subsystems monitored by related sensor classes, making them well suited to a foundation-model paradigm that learns transferable representations from operational data.

**Scientific foundation models.** The foundation-model paradigm has been established in language modeling (Devlin et al., 2019; Radford et al., 2019) and computer vision (He et al., 2022; Caron et al., 2021), and has recently gained momentum across scientific domains, including physical systems (Rautela et al., 2025; Park et al., 2025; Nguyen et al., 2023), chemical science (Wadell et al., 2025), biological science (Moor et al., 2023; Brixi et al., 2026), and time-series prediction (Ansari et al., 2024; Nie et al., 2022). Across these domains, foundation models derive their strength from self-supervised pretraining on large, heterogeneous datasets together with flexible input representations. Accelerator telemetry naturally exhibits these same characteristics: it is abundant but weakly labeled, heterogeneous across subsystems and facilities, and often noisy, incomplete, and affected by changing operating conditions.

## 3. Experiments

### 3.1. Datasets

The datasets used in this study span diverse accelerator facilities, sensor modalities, and operational tasks. As summarized in Table 4, they include multi-subsystem operational measurements from LANSCE at LANL (Scheinker, 2025), beam-loss measurements from LHC at CERN (Krymova et al., 2023), high-voltage converter modulator (HVCM) waveforms for power-electronics fault prognosis at SNS at ORNL (Radaideh et al., 2023a;b), superconducting radio-frequency (SRF) cavity fault data from CEBAF at JLab (Tennant et al., 2020), and RF-station anomaly data from LCLS at SLAC (Humble et al., 2022; Liang et al., 2025). Together, these datasets provide a heterogeneous benchmark for evaluating whether a single self-supervised sensor model can generalize across facilities, subsystems, sampling structures, and downstream tasks.

### 3.2. Main results

For readability and conciseness, we defer details of the experimental settings, including model variants, windowing, masking, normalization, noise augmentation, the training objective, optimization, and compute, to Appendix C.

**Scaling study.** A detailed *scaling study* in Appendix D shows that validation masked-reconstruction loss decreases with both model size and pretraining data volume, with the best fitted point occurring at the largest evaluated model–data scale. In our setting, the fitted exponents are $\alpha = 0.715$ for model size and $\beta = 0.374$ for data size, indicating that

*Table 1.* Zero-shot imputation performance on held-out test sets, reported as raw (un-normalized scale) mean absolute error (MAE) averaged over masked entries for all samples in the test set. *Lower values indicate better reconstruction.* Bold values denote the best model for each dataset and mask ratio.

| Dataset | Mask ratio | XTi | Ti | Mi | S | B |
|---|---|---|---|---|---|---|
| LANSCE 2019–2024 | 5% | 2.4928 | 1.9969 | 2.0296 | 1.8414 | **1.7130** |
| | 15% | 2.4693 | 1.9984 | 1.9526 | 1.8916 | **1.6769** |
| | 25% | 2.5411 | 2.1002 | 1.9933 | 2.0337 | **1.7580** |
| | 50% | 2.6005 | 2.3238 | 2.1687 | 2.1835 | **2.0624** |
| LANSCE-2025 | 5% | 16.4059 | 11.0818 | 6.7363 | 6.8370 | **6.1174** |
| | 15% | 8.4765 | 7.2481 | 7.7511 | 7.3025 | **6.6987** |
| | 25% | 7.9151 | 7.3074 | 8.1885 | 7.8608 | **6.9565** |
| | 50% | 8.4266 | 8.1807 | 7.9797 | **7.4946** | 8.6389 |
| LHC-BL | 5% | 3.0867 | 2.8527 | 2.6608 | 2.4952 | **2.3110** |
| | 15% | 3.1752 | 2.9858 | 2.8162 | 2.5671 | **2.5179** |
| | 25% | 3.1206 | 3.1030 | 2.9639 | **2.5627** | 2.5879 |
| | 50% | 2.9752 | 2.9677 | 3.0199 | 2.8093 | **2.7347** |
| HVCM-SNS | 5% | 136.5608 | 166.7894 | 134.7735 | 130.5587 | **127.9945** |
| | 15% | 160.5342 | 157.6206 | 151.5387 | **139.9381** | 140.3383 |
| | 25% | 183.9417 | 189.1077 | 177.0788 | 170.1272 | **167.9301** |
| | 50% | 201.4619 | 219.4875 | 198.7487 | 207.8971 | **197.6245** |
| RFS-LCLS | 5% | 0.9644 | 0.8713 | 0.8095 | 0.7624 | **0.7234** |
| | 15% | 0.9726 | 0.8775 | 0.8295 | 0.7489 | **0.7220** |
| | 25% | 1.0196 | 0.9111 | 0.8763 | 0.8017 | **0.7630** |
| | 50% | 1.1120 | 1.0106 | 1.0497 | 0.8997 | **0.8896** |
| SRF-CEBAF | 5% | 0.2306 | 0.2288 | 0.2347 | 0.2177 | **0.2099** |
| | 15% | 0.2268 | 0.2278 | 0.2264 | 0.2197 | **0.2123** |
| | 25% | 0.2241 | 0.2220 | 0.2228 | 0.2194 | **0.2140** |
| | 50% | 0.2247 | 0.2182 | 0.2248 | 0.2208 | **0.2137** |

*Table 2.* All ten evaluated models, including nine prior models and SensOFormer-XTi, achieved perfect precision, recall, F1, and AUC. The main comparison uses the 21 induced early-fault tests, where a test is counted as passed when $\rho < 25\%$ and F1 $> 0.8$, following (Radaideh et al., 2023b). VE = hierarchical voting ensemble, AB = AdaBoost, ET = extremely randomized trees, CNN = convolutional neural network, BC = bagging classifier, RF = random forest, GB = gradient boosting, LSVM = linear support vector machine, RBF-SVM = radial-basis-function support vector machine, and SOF-XTi = SensOFormer-XTi.

| Metric | VE | AB | ET | CNN | BC | RF | GB | LSVM | RBF-SVM | SOF-XTi |
|---|---|---|---|---|---|---|---|---|---|---|
| Passed tests | **20/21** | 11/21 | 10/21 | 9/21 | 9/21 | 9/21 | 9/21 | 5/21 | 5/21 | **20/21** |
| Success rate (%) | **95.2** | 52.4 | 47.6 | 42.9 | 42.9 | 42.9 | 42.9 | 23.8 | 23.8 | **95.2** |

both larger models and larger pretraining corpora improve reconstruction loss within the evaluated regime. The resulting compute-optimal trend allocates additional compute to both dimensions, with data size increasing faster than parameter count, suggesting that future accelerator foundation models should scale the pretraining corpus alongside model capacity.

**Zero-shot missing-data imputation.** Table 1 reports zero-shot point-wise imputation performance across datasets, mask ratios, and model variants: XTi (2.38M), Ti (5.80M), Mi (13.12M), S (35.87M), and B (80.54M). Lower values indicate better reconstruction. Overall, the B model achieves the lowest error in 21 of the 24 dataset–mask combinations, while the S model is best in the remaining three settings. This shows a clear benefit from increasing model

capacity for zero-shot imputation, suggesting that larger SensOFormer variants more effectively use temporal and cross-channel context to reconstruct missing sensor values. The LANSCE-2025 setting is more challenging than the LANSCE 2019–2024 held-out test set because it evaluates transfer to a different operating year rather than to a held-out test set from the same pretraining-year distribution. The reconstruction errors are substantially higher on LANSCE-2025, indicating a temporal distribution shift in the sensor statistics, operating conditions, or available channel coverage.

**Qualitative reconstructions.** Figs. 4 – 12 (Appendix E) further support the quantitative results of Table 1. The predicted values at masked locations closely follow the normalized ground-truth trajectories in representative held-

out test-set windows, showing that the model is not only reduces aggregate MAE but also recovers local temporal structure. Notably, this behavior is also observed on the LANSCE-2025 dataset (Figs. 6 and 7), indicating that the pretrained model retains useful reconstruction ability under distribution shift. Together, the quantitative and qualitative results demonstrate that masked-denoising pretraining learns reusable sensor representations that can be applied directly to missing-data imputation across heterogeneous accelerator datasets.

**Anomaly detection.** We formulate supervised anomaly detection as a binary classification problem on the HVCM-SNS dataset. The labeled dataset contains 10,000 waveform samples, split into 8,000 training samples and 2,000 held-out test samples, and the operationally relevant evaluation is performed on 21 induced early-fault test files following the protocol of Radaideh et al. (2023b). Each input consists of multichannel HVCM waveforms, and the model predicts whether the signal corresponds to normal operation or a faulty condition. Additional experimental details are provided in Appendix C. On the 2,000-sample test split, all methods, including SensOFormer, achieve perfect precision, recall, F1, and AUC. As shown in Table 2, SensOFormer-XTi passes 20 of the 21 early-fault tests, matching the hierarchical voting ensemble and substantially outperforming the individual classical and CNN baselines under this early-detection criterion.

**Fault & cavity identification.** We evaluate SensOFormer-XTi on supervised SRF fault diagnosis using the SRF-CEBAF dataset. The CEBAF C100 cryomodule data support two related multiclass classification tasks: *cavity identification*, where the goal is to identify the first faulting cavity, and *fault identification*, where the goal is to classify the fault type. These tasks are challenging because SRF cavities in a cryomodule are strongly coupled, so a primary fault can quickly induce secondary responses in neighboring cavities. Additional experimental details are provided in Appendix C. Table 3 compares SensOFormer-XTi with prior deep-learning baselines on the CEBAF test set. For cavity identification, SensOFormer-XTi achieves 84.80% accuracy, comparable to the best prior LSTM and CNN direct-mapping baselines. For fault identification, SensOFormer-XTi achieves 84.01% accuracy, substantially improving over the prior 74.10% baselines.

**Data-scarce fault identification.** We further evaluate fault identification in a data-scarce setting, motivated by the fact that accelerator fault data are costly to collect and require expert annotation. To assess label efficiency, we vary the fraction of labeled training examples while keeping the test set fixed. As shown in Figure 1, SensOFormer-XTi improves monotonically as additional labeled data are in-

*Table 3.* Test-set accuracy for CEBAF C100 cavity identification and fault identification. We compare SensOFormer-XTi with prior deep-learning baselines from Shabalina et al. (2020). SensOFormer-XTi matches the strongest baselines on cavity identification and improves fault identification accuracy from 74.10% to 84.01%.

| Task | Method | Accuracy (%) |
|---|---|---|
| Cavity identification | LSTM | 84.60 |
| | CNN, direct | 84.60 |
| | CNN, scalogram | 45.00 |
| | **SOF-XTi (Ours)** | **84.80** |
| Fault identification | LSTM | 74.10 |
| | CNN, direct mapping | 74.10 |
| | CNN, scalogram | 58.90 |
| | **SOF-XTi (Ours)** | **84.01** |

troduced and surpasses the prior deep-learning benchmark using only 25% of the training set. This result suggests that self-supervised pretraining provides a useful representation for SRF fault diagnosis, reducing the amount of labeled fault data needed to achieve competitive performance.

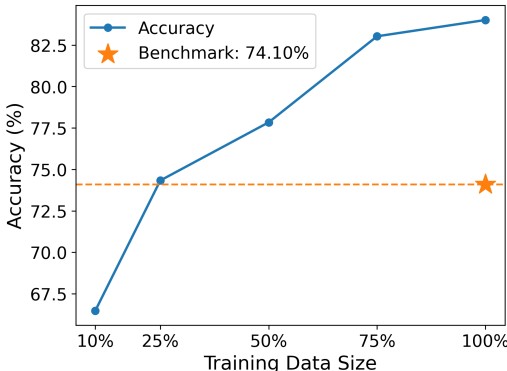

*Figure 1.* Fault-identification accuracy vs training-data fraction.

## 4. Conclusion

We introduced SensOFormer, a self-supervised foundation model for particle-accelerator operational data. SensO-Former learns temporal and cross-channel structure from heterogeneous and noisy measurements. Across datasets, SensOFormer demonstrates strong transfer to missing-data reconstruction, anomaly detection, cavity identification, and fault identification. The scaling study further shows that reconstruction performance improves with both model size and pretraining data, suggesting that accelerator foundation models can benefit from systematic scaling. Overall, SensOFormer provides a reusable foundation-model backbone for accelerator operational data, reducing the need to train separate models for each subsystem or diagnostic task.

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

## A. Datasets

Table 4 summarizes the tensorized form of each dataset and the downstream tasks used in our evaluation. The datasets differ substantially in both temporal length and channel structure: LANSCE contains variable-channel operational telemetry, LHC-BL provides very long beam-loss time series, LCLS and HVCM-SNS contain fixed-channel waveform-style records, and SRF-CEBAF consists of short RF fault-event windows. All datasets are used for missing-data imputation (MDI), while HVCM-SNS and SRF-CEBAF additionally support diagnostic tasks such as anomaly detection, cavity identification, and fault identification. This design allows us to evaluate both zero-shot reconstruction across heterogeneous sensor formats and transfer to task-specific accelerator diagnostics.

*Table 4.* Datasets used for pretraining and downstream evaluation. Tensor shapes are reported as $(N, T, C)$, where $N$ is the number of examples, $T$ is the number of time steps, and $C$ is the number of sensors/channels. MDI denotes missing-data imputation, and $N^*$ denotes a variable number of samples and $C^*$ denotes a variable number of channels, with the bracketed range indicating the minimum and maximum number of channels across the dataset.

| Dataset | Facility / subsystem | Dataset/Tensor shape $(N,T,C)$ | Downstream task |
|---|---|---|---|
| LANSCE (2019–2024) | LANSCE operational data | $(4250, 10000, C^* \in [1, 1754])$ | MDI on 10% test set |
| LANSCE (2025) | LANSCE operational data | $(17, 10000, C^* \in [1, 769])$ | MDI on 100% test set |
| LHC-BL (2017) LHC-BL (2018) | LHC beam-loss monitoring | $(1, 542061, 66)$ $(1, 524430, 66)$ | MDI on 10% test set |
| RFS-LCLS (2020) | LCLS RF station signals (AMM and beam) LCLS RF station signals (AMPL and beam) | $(1269, 2196, 9)$ $(3208, 2076, 9)$ | MDI on 10% test set |
| RFS-LCLS (2024) | LCLS RF station signals (phase and beam) | $(5364, 1066, 9)$ | MDI on 10% test set |
| HVCM-SNS | SNS high-voltage converter modulators | $(10000, 3753, 12)$ $(N^*, 3753, 12)$ | 1. MDI on 10% test set 2. Anomaly detection |
| SRF-CEBAF | CEBAF superconducting RF cavities | $(2375, 192, 1)$ | 1. MDI on 10% test set 2. Cavity identification 3. Fault identification |

# B. Proposed Method

## B.1. Pretraining strategy

Before describing the model architecture, we first motivate the pretraining objective. Foundation models are commonly pretrained using either autoregressive prediction, such as next-token, next-step, or next-frame prediction (Radford et al., 2019; Ansari et al., 2024; Rautela et al., 2025), or masked reconstruction and denoising objectives (Devlin et al., 2019; Nie et al., 2022; Wadell et al., 2025). Autoregressive forecasting is a natural pretraining choice when the primary goal is to extrapolate a sequence forward in time, as in language modeling, time-series forecasting for financial, energy, or traffic data, and physical systems with temporally evolving fields such as atmosphere and climate. Accelerator operational telemetry, however, does not generally form a single uniform sequence governed by one dominant time scale. Beam-related signals can correspond to fast machine events, since charged-particle bunches travel near the speed of light, whereas other subsystems, such as cooling, vacuum, and power-supply systems, evolve over slower operational time scales. Moreover, a central downstream task in this work is to recover missing or corrupted sensor measurements, rather than only to extrapolate future time steps. These considerations make masked denoising reconstruction a more natural pretraining objective for accelerator telemetry: the model must infer hidden measurements from the available temporal and cross-channel context while learning representations robust to noise, missingness, and heterogeneous sensor structure.

We therefore pretrain SensOFormer using a denoising masked reconstruction objective, illustrated in Figure 2(a). For each training instance, we sample a corruption pattern from a mask bank and hide the corresponding subset of sensor measurements. The mask bank includes point-wise, time-wise, channel-wise, and block-wise masks, which emulate common failure modes in accelerator telemetry such as isolated dropouts, missing time intervals, unavailable sensor channels, and structured channel–time gaps. The visible entries are additionally perturbed with additive noise sampled from multiple noise strengths, including zero, while the mask indicator specifies which entries have been hidden. Given a corrupted input batch and the corresponding binary mask, SensOFormer is trained to reconstruct the original sensor values at the masked locations. By combining structured masking with denoising, the pretraining task discourages simple copying of clean observations and instead encourages the model to learn robust temporal and cross-channel structure from noisy, incomplete accelerator data.

## B.2. Overview of SensOFormer

Figure 2(b) illustrates SensOFormer as an encoder–processor–decoder architecture for accelerator sensor sets. The key design choice is to represent the sensors observed at each time step as an unordered set rather than as a fixed channel vector. This allows the model to handle variable sensor availability, channel ordering, and subsystem-specific measurements without tying the representation to a particular global sensor layout. Given a corrupted input batch and its mask indicator, SensOFormer forms per-channel features $x \in \mathbb{R}^{B \times L \times C \times F}$, where $B$ is the batch size, $L$ is the sequence length, $C$ is the number of sensor channels, and $F$ is the feature dimension. In our implementation, $F = 2$, corresponding to the corrupted sensor value and a binary mask indicator. The batch and time dimensions are flattened to obtain $(BL, C, F)$, so that each time step is treated as a set of $C$ channel observations. A shared tokenization layer maps these features to channel tokens in $\mathbb{R}^{BL \times C \times d}$.

The *set encoder* uses a Perceiver-style cross-attention mechanism in which $S$ learnable latent queries attend to the channel tokens. Because the same attention operation is applied to an unordered set of sensor tokens, the encoder is permutation-invariant with respect to channel ordering: reordering the input sensors does not change the aggregated latent representation. The resulting slot representations are pooled and reshaped into a latent sequence $z \in \mathbb{R}^{B \times L \times d}$, which is processed by a transformer to model temporal dependencies in a compact shared latent space.

The *set decoder* maps the processed latent sequence back to channel-wise predictions. For each latent vector $z_{b,t}$, the decoder constructs $M$ memory slots, and the channel tokens are reused as output queries that cross-attend to these memory slots. This produces one prediction per queried channel, followed by layer normalization and scalar projection to obtain $\hat{x} \in \mathbb{R}^{B \times L \times C}$. The decoder is permutation-equivariant: reordering the output channel queries reorders the corresponding predictions in the same way. Thus, SensOFormer combines permutation-invariant encoding, temporal latent processing, and permutation-equivariant decoding to model variable sensor sets without requiring a fixed channel layout.

## B.3. Downstream tasks

Figure 2(c) summarizes how SensOFormer is adapted to downstream tasks after pretraining. We consider three task instantiations: missing-data imputation, anomaly detection, and fault identification. In all cases, the set encoder $E$ and

## a) Masked Denoising strategy (pretraining)

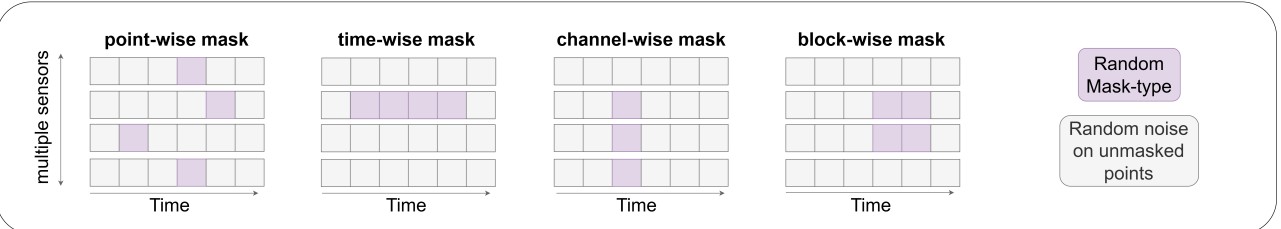

## b) SensOFormer (Set Encoder, latent transformer/processor and Set Decoder)

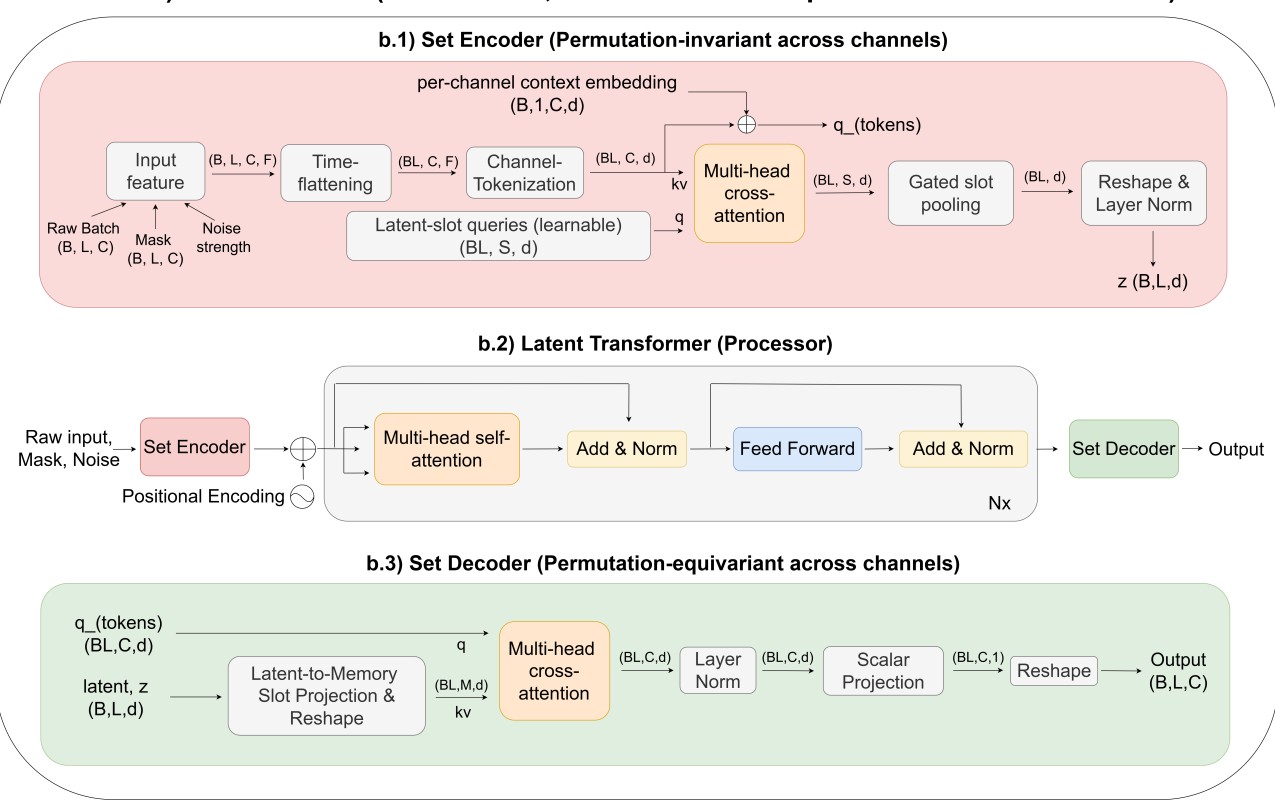

## c) Downstream tasks

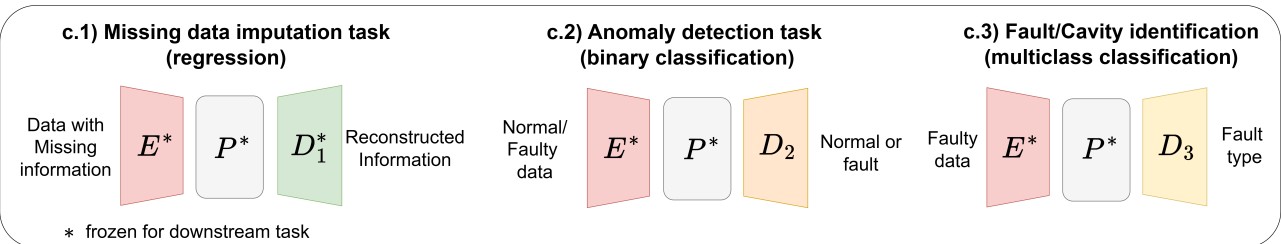

*Figure 2.* **Overview of SensOFormer.** SensOFormer uses an encoder–processor–decoder architecture for variable sensor sets. The perceiver-style set encoder is permutation-invariant across sensor channels. The latent transformer processor models temporal dependencies. The set decoder is permutation-equivariant, producing channel-wise reconstructions that preserve the queried sensor identities.

transformer processor $P$ provide a shared representation of the operational sensor data, while the task-specific decoder or head $D_i$ determines the output type. This design allows the same pretrained backbone to support reconstruction, binary classification, and multiclass classification tasks.

For *missing-data imputation*, the input is partially observed telemetry in which missingness can occur as missing sensor values, consecutive missing time steps, missing channel values over multiple time steps, or structured channel–time blocks. Simple interpolation or local averaging can be effective for isolated missing points, but becomes unreliable when multiple time windows or sensor channels are absent. As shown in Fig. 2(c.1), SensOFormer uses the reconstruction decoder $D_1$ to predict the missing values from the available temporal and cross-channel context. This task is directly aligned with the denoising masked reconstruction objective used during pretraining, so the architecture can be reused for missing data reconstruction.

For *anomaly detection*, the goal is to distinguish normal operation from abnormal or faulty machine behavior using operational sensor data. This task is challenging because labeled anomalies are limited: fault annotation requires domain expertise and careful post-hoc analysis. Reliable anomaly detection is operationally important because accelerator facilities run under scheduled beam-time periods and can be interrupted by planned or unplanned maintenance. As shown in Fig. 2(c.2), we adapt the pretrained encoder–processor backbone with a binary classification head $D_2$ to identify normal versus faulty operating patterns.

For *cavity and fault identification*, the objective is to classify the type or source of a fault once faulty data are available. This is a more detailed multiclass classification problem than binary anomaly detection, requiring the model to distinguish between different fault categories or faulty components. As shown in Fig. 2(c.3), we reuse the same pretrained encoder $E$ and processor $P$, and attach a task-specific multiclass head $D_3$ to predict the fault (and cavity) labels.

## C. Experimental settings

**Model variants.** We evaluate five SensOFormer variants, summarized in Table 5. The variants differ in the latent attention dimension, number of attention heads, and transformer depth, while keeping the same encoder–processor–decoder architecture. For all variants, the MLP dimension is set to $4d$, where $d$ is the attention dimension. Unless otherwise specified, the set encoder uses 16 latent slots, the set decoder uses 32 memory slots, gated pooling is used in the set encoder, and the transformer uses absolute positional encodings with maximum length 1024. We restrict the largest model to the B-scale variant based on the scaling study in Appendix D, where the performance gains begin to show diminishing returns relative to the additional training cost.

*Table 5.* SensOFormer model variants used in this study. The MLP dimension is set to $4d$ for all variants, where $d$ is the attention dimension.

| Model | Attention dim. $d$ | Heads | Depth | MLP dim. | Parameters |
|-------|--------------------|-------|-------|----------|------------|
| XTi | 192 | 4 | 2 | 768 | 2.38M |
| Ti | 256 | 4 | 4 | 1024 | 5.80M |
| Mi | 384 | 6 | 4 | 1536 | 13.12M |
| S | 512 | 8 | 8 | 2048 | 35.87M |
| B | 768 | 12 | 8 | 3072 | 80.54M |

**Pretraining data and windowing.** SensOFormer is pretrained on the heterogeneous accelerator datasets described in Table 4, including LANSCE, LHC-BL, HVCM-SNS, RFS-LCLS, and SRF-CEBAF data. Each original time series is treated as a separate sequence, and the train/validation split is performed at the sequence level before windowing. This prevents overlapping windows from the same original sequence from appearing in both training and validation sets. Unless otherwise specified, 90% of the available sequences are used for pretraining and 10% are used for validation. Each sequence is then segmented into overlapping windows of length $L = 1024$ with stride 256. The number of sensor channels is kept dataset-specific, and the model is trained without padding along the channel dimension. This is possible because the set encoder and set decoder operate on sensor sets rather than on a fixed global channel layout.

**Masking.** For each training window, we uniformly sample one mask type from a mask bank and train the model to reconstruct the held-out entries from the remaining visible measurements. The mask bank contains four masking patterns: point-wise, time-wise, channel-wise, and block-wise masking. Point-wise masking independently masks entries with probability 0.15. Time-wise masking samples one time index and masks a fraction 0.75 of the channels at that time index. Channel-wise masking samples one sensor channel and masks a fraction 0.60 of its time steps. Block-wise masking samples a contiguous rectangular region in the time–channel plane with area approximately $0.15LC$, where $L$ is the window length and $C$ is the number of channels; the block aspect ratio is sampled randomly. For the time-wise, channel-wise, and block-wise masks, we avoid masking an entire row, column, or window whenever possible. These patterns emulate common missingness modes in accelerator telemetry, including isolated dropouts, partial time-slice loss, intermittent sensor-channel loss, and structured channel–time gaps. Masked entries are replaced by a sentinel value, while the binary mask is provided as an input feature so that the model can distinguish intentionally hidden entries from valid observed measurements.

**Normalization.** Because the pretraining corpus combines sensors with different physical units, dynamic ranges, and operating regimes, we normalize each window on the fly rather than using global dataset statistics. Normalization is performed independently for each batch element and sensor channel, using only the visible entries in that window. Let $x_{b,t,c}$ denote the value for batch element $b$, time index $t$, and channel $c$, and let $M_{b,t,c} = 1$ indicate a masked target. For each $(b, c)$ pair, we compute

$$x_{b,c}^{\min} = \min_{t:M_{b,t,c}=0} x_{b,t,c}, \qquad x_{b,c}^{\max} = \max_{t:M_{b,t,c}=0} x_{b,t,c}, \tag{1}$$

and define the normalization range as

$$r_{b,c} = \max\left(x_{b,c}^{\max} - x_{b,c}^{\min}, \epsilon\right), \tag{2}$$

with $\epsilon = 10^{-5}$. The normalized signal is then

$$\tilde{x}_{b,t,c} = \text{clip}\left(\frac{x_{b,t,c} - x_{b,c}^{\min}}{r_{b,c}}, 0, 1\right). \tag{3}$$

Computing the statistics only from unmasked entries prevents information from the held-out targets from leaking into the input normalization. This per-window, per-channel normalization removes dataset-specific scale differences while preserving the relative temporal structure within each sensor. The same masked normalization procedure is used during validation and downstream imputation.

**Noise augmentation.**    After normalization, we apply denoising augmentation during pretraining. For each training batch, we sample a noise level

$$\sigma \in \{0,\ 0.005,\ 0.01,\ 0.015\}, \tag{4}$$

and add zero-mean Gaussian noise only to the visible entries:

$$\tilde{x}^{\text{in}}_{b,t,c} = \tilde{x}_{b,t,c} + (1 - M_{b,t,c})\epsilon_{b,t,c}, \qquad \epsilon_{b,t,c} \sim \mathcal{N}(0, \sigma^2). \tag{5}$$

Masked entries are not perturbed by this noise process; they are represented to the model through the mask indicator and sentinel value. Including $\sigma = 0$ keeps a subset of batches noise-free, while the nonzero noise levels encourage the model to learn robust denoising representations instead of simply copying the visible measurements. Noise augmentation is used only during training and is not applied during validation or zero-shot imputation.

**Training objective.**    The model is trained to reconstruct the clean normalized signal from the corrupted input. Let $x$ be the normalized target, $\hat{x}$ be the model prediction, and $M$ be the binary mask, where $M_{t,c} = 1$ indicates a masked entry. The training loss is

$$\mathcal{L} = \frac{\sum_{t,c} M_{t,c}(\hat{x}_{t,c} - x_{t,c})^2}{\sum_{t,c} M_{t,c}} + \lambda_{\text{unmask}} \frac{\sum_{t,c}(1 - M_{t,c})(\hat{x}_{t,c} - x_{t,c})^2}{\sum_{t,c}(1 - M_{t,c})}$$
$$+ \lambda_{\text{smooth}} \frac{1}{(L-1)C} \sum_{t=1}^{L-1} \sum_{c=1}^{C} |\hat{x}_{t+1,c} - \hat{x}_{t,c}|. \tag{6}$$

The first term is the masked reconstruction loss and is the primary pretraining objective. The second term weakly regularizes reconstruction on unmasked entries, and the third term penalizes high-frequency variation in the predicted sequence. We use $\lambda_{\text{unmask}} = 0.05$ and $\lambda_{\text{smooth}} = 0.01$. Validation loss is computed only on masked entries, matching the missing-data imputation objective.

**Optimization and compute.**    Pretraining is performed using distributed data parallelism on 16 NVIDIA H100 GPUs with 80GB memory each. Each GPU processes one micro-batch, and gradient accumulation is used before applying an optimizer update. With 16 GPUs, a per-GPU micro-batch size of 1, and accumulation over 4 local micro-batches per GPU, the effective global batch size is $16 \times 1 \times 4 = 64$. All SensOFormer variants are pretrained for 25 epochs, where one epoch denotes one pass through the pretraining set. This corresponds to approximately 125K optimizer updates over the full pretraining run. We use the same optimization recipe for all model sizes so that performance differences are primarily attributable to model scale. We use AdamW with learning rate of $10^{-5}$ and the default PyTorch weight decay. The learning rate follows a warmup-cosine schedule: the first 2.5% of optimizer updates are used for linear warmup, followed by cosine decay over the remaining updates. We use automatic mixed precision with bfloat16 to reduce memory usage and improve throughput, while computing the reconstruction loss in full precision for numerical stability. Gradients are clipped to a maximum norm of 1.0 before each optimizer update. All downstream inference and supervised adaptation experiments were performed on a single NVIDIA RTX A6000 GPU with 48 GB of memory.

**Fine-tuning settings: Anomaly detection.**    To adapt SensOFormer-XTi to this task, we freeze the pretrained set encoder and latent transformer processor and train only a lightweight binary classification head on top of the latent sequence representation. The head is a one-dimensional temporal CNN that operates on the latent sequence $z \in \mathbb{R}^{B \times L \times d}$ with four Conv1D blocks with GELU activations and max pooling after each convolution. The resulting temporal features are summarized using both global max pooling and global mean pooling, concatenated, normalized, and passed through a three-layer MLP before producing a single binary logit. The classification head contains 0.22M trainable parameters. We train the detector head for 200 epochs using AdamW with a learning rate of $10^{-5}$.

**Fine-tuning settings: Cavity and fault identification.** To adapt SensOFormer-XTi to these tasks, we reuse the pretrained encoder–processor backbone and replace the reconstruction decoder with a lightweight temporal CNN classification head. For each event, the backbone produces a latent sequence $z \in \mathbb{R}^{B \times 192 \times d}$, where 192 is the event-window length and $d$ is the latent feature dimension. We train separate heads for cavity identification and fault identification, with output dimensions of 9 and 8 classes, respectively. The head first applies layer normalization to the latent sequence, then processes it with two one-dimensional convolutional layers along the temporal dimension, followed by three fully connected layers. This task-specific head contains approximately $0.5$M trainable parameters. We train for 200 epochs using AdamW with a learning rate of $10^{-4}$ and batch size of 32.

# D. Scaling studies

Scaling studies are important in the foundation-model setting because they quantify how pretraining loss changes as a function of model size, data size, and training compute. In language modeling, prior work has shown that loss often follows smooth power-law trends with the number of model parameters, training tokens, and compute (Kaplan et al., 2020). The Chinchilla study further showed that compute-efficient training requires scaling model size and data size together, rather than increasing model capacity alone (Hoffmann et al., 2022). We perform an analogous scaling analysis for SensOFormer to understand how masked-reconstruction loss depends on the number of model parameters and the amount of pretraining data.

We fit the validation masked-reconstruction loss using a separable scaling law:

$$\mathcal{L}(N, D) = \mathcal{L}_\infty + \frac{A}{N^\alpha} + \frac{B}{D^\beta}, \tag{7}$$

where $N$ is the number of trainable model parameters, $D$ is the amount of pretraining data, and $\mathcal{L}(N, D)$ is the validation loss. The constant $\mathcal{L}_\infty$ represents the asymptotic loss approached by the fitted model in the limit of large $N$ and $D$. The terms $A/N^\alpha$ and $B/D^\beta$ represent model-limited and data-limited contributions to the loss, respectively. The exponents $\alpha$ and $\beta$ determine how quickly each contribution decreases as model size or data size increases.

We use this scaling analysis as an empirical diagnostic rather than as an extrapolation guarantee. SensOFormer differs from language models in both data structure and objective: the inputs are heterogeneous accelerator sensor windows rather than text tokens, and the pretraining task is masked denoising reconstruction rather than next-token prediction. Nevertheless, Eq. (7) provides a compact way to summarize whether increasing model capacity and pretraining data improves learned accelerator representations. In our study, we vary $N$ using the five SensOFormer variants in Table 5 and vary $D$ by subsampling the pretraining corpus at 12.5%, 25%, 50%, and 100%. Each scaling run uses single-pass pretraining, and the loss is evaluated on the validation set. The scaling study is performed with an effective batch size of 64 on 32 NVIDIA H100 GPUs.

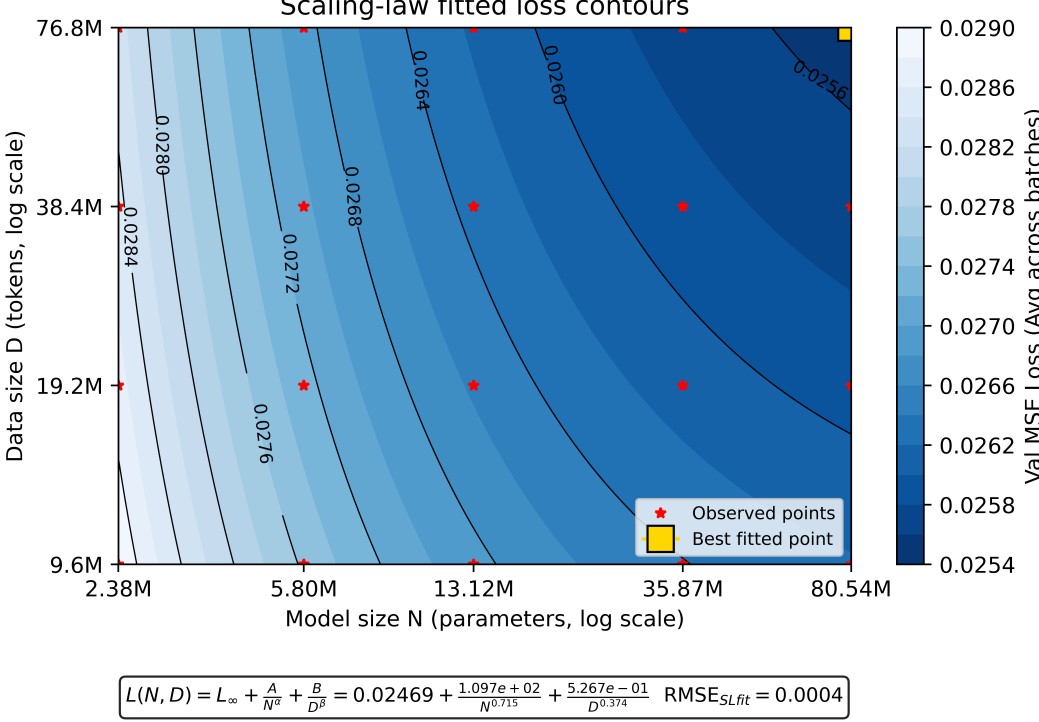

*Figure 3.* Scaling-law fit for SensOFormer validation masked-reconstruction loss as a function of model size $N$ and pretraining data size $D$. Red markers denote observed model–data configurations, and the yellow marker denotes the best fitted point within the evaluated grid. The fitted surface shows decreasing loss with both model capacity and data size.

Figure 3 shows the fitted scaling surface obtained by nonlinear least squares. The fitted loss is

$$\mathcal{L}(N, D) = 0.02469 + \frac{1.097 \times 10^2}{N^{0.715}} + \frac{5.267 \times 10^{-1}}{D^{0.374}}, \tag{8}$$

with an RMSE of $4 \times 10^{-4}$ over the observed model–data configurations. Here, $N$ and $D$ are measured in raw counts rather than in millions. The low RMSE indicates that the separable power law fits the measured validation losses well within the evaluated grid. We do not interpret this as a guarantee that the same law will hold outside the measured range.

The fitted surface shows that validation loss decreases as both model size and data size increase. The best fitted point lies at the largest evaluated model and data scale, indicating that SensOFormer continues to benefit from scale in this regime. The contour geometry also shows that increasing model size produces a strong loss reduction across the evaluated range, especially when moving from the smallest models to the larger S and B variants. Increasing data size also improves loss, but the fitted exponent for data is smaller than that for model size, with $\beta = 0.374$ and $\alpha = 0.715$. This means that, within the fitted law, the data-limited term decays more slowly with additional data than the model-limited term decays with additional parameters. Thus, although the current evaluated regime still benefits strongly from increasing model capacity, future compute-efficient scaling should increase both model size and pretraining data rather than scaling parameters alone.

We can use the fitted law to derive the compute-optimal allocation of model size and data size (Hoffmann et al., 2022). For a fixed training recipe, let the training compute budget be approximately proportional to the product of model size and data size,

$$\mathcal{C} = \kappa N D, \tag{9}$$

where $\kappa$ absorbs architecture- and implementation-dependent constants. Let $K = \mathcal{C}/\kappa$, so the compute constraint is $ND = K$. Ignoring the constant $\mathcal{L}_\infty$, the compute-optimal allocation solves

$$\min_{N,D} \left( \frac{A}{N^\alpha} + \frac{B}{D^\beta} \right) \quad \text{s.t.} \quad ND = K. \tag{10}$$

Substituting $D = K/N$ and differentiating with respect to $N$ gives the optimality condition

$$\alpha A (N^\star)^{-\alpha} = \beta B (D^\star)^{-\beta}. \tag{11}$$

This condition states that, at the compute-optimal point, the marginal benefit of increasing model size is balanced against the marginal benefit of increasing data size under the fixed-compute constraint. Solving for the optimal model and data scales gives

$$N^\star(K) = \left( \frac{\alpha A}{\beta B} \right)^{\frac{1}{\alpha+\beta}} K^{\frac{\beta}{\alpha+\beta}}, \qquad D^\star(K) = \left( \frac{\beta B}{\alpha A} \right)^{\frac{1}{\alpha+\beta}} K^{\frac{\alpha}{\alpha+\beta}}. \tag{12}$$

Using the fitted exponents from Eq. (8), we obtain

$$N^\star \propto \mathcal{C}^{0.343}, \qquad D^\star \propto \mathcal{C}^{0.657}. \tag{13}$$

Under the fitted scaling law, additional compute should therefore be allocated to both larger models and more pretraining data. However, the compute-optimal trajectory increases data size faster than parameter count as the compute budget grows. This is consistent with the broader lesson from Chinchilla-style scaling: larger models are most useful when accompanied by sufficient pretraining data. For SensOFormer, the scaling study suggests that future accelerator foundation models should expand the pretraining corpus alongside model capacity to remain compute-efficient.

# E. Extended Results

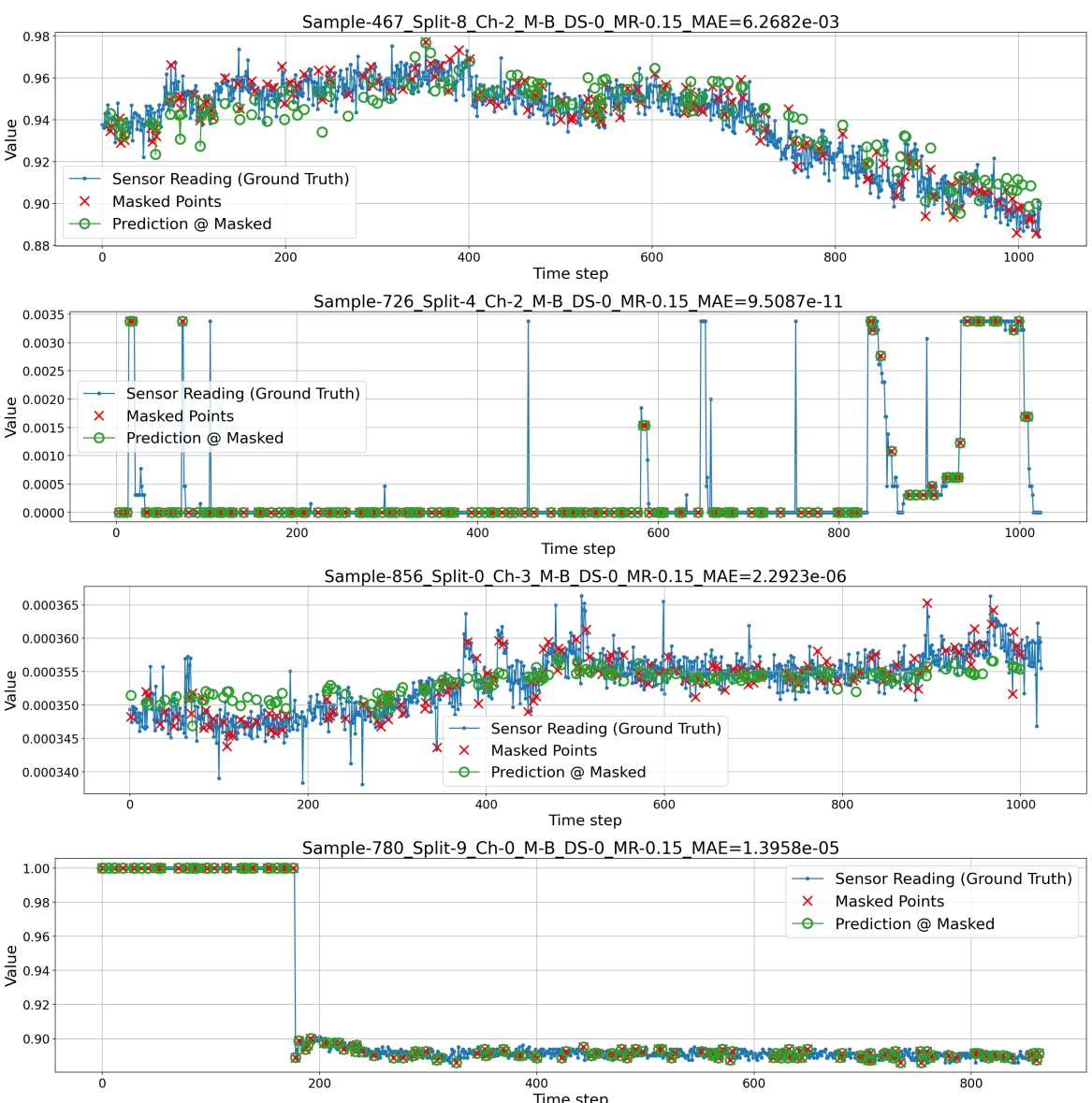

*Figure 4.* Representative zero-shot *point-wise imputation* examples from pretrained *SensOFormer-B* on held-out *LANSCE 2019–2024* test windows with a *15% mask ratio*. Blue curves show the normalized ground-truth sensor values, red markers indicate masked entries, and green circles show the model predictions at the masked locations. Each panel corresponds to a different test window and channel. The subplot title reports the sample index, split, channel, model size (M), dataset (DS), mask ratio (MR), and mean absolute error (MAE) computed on the masked entries.

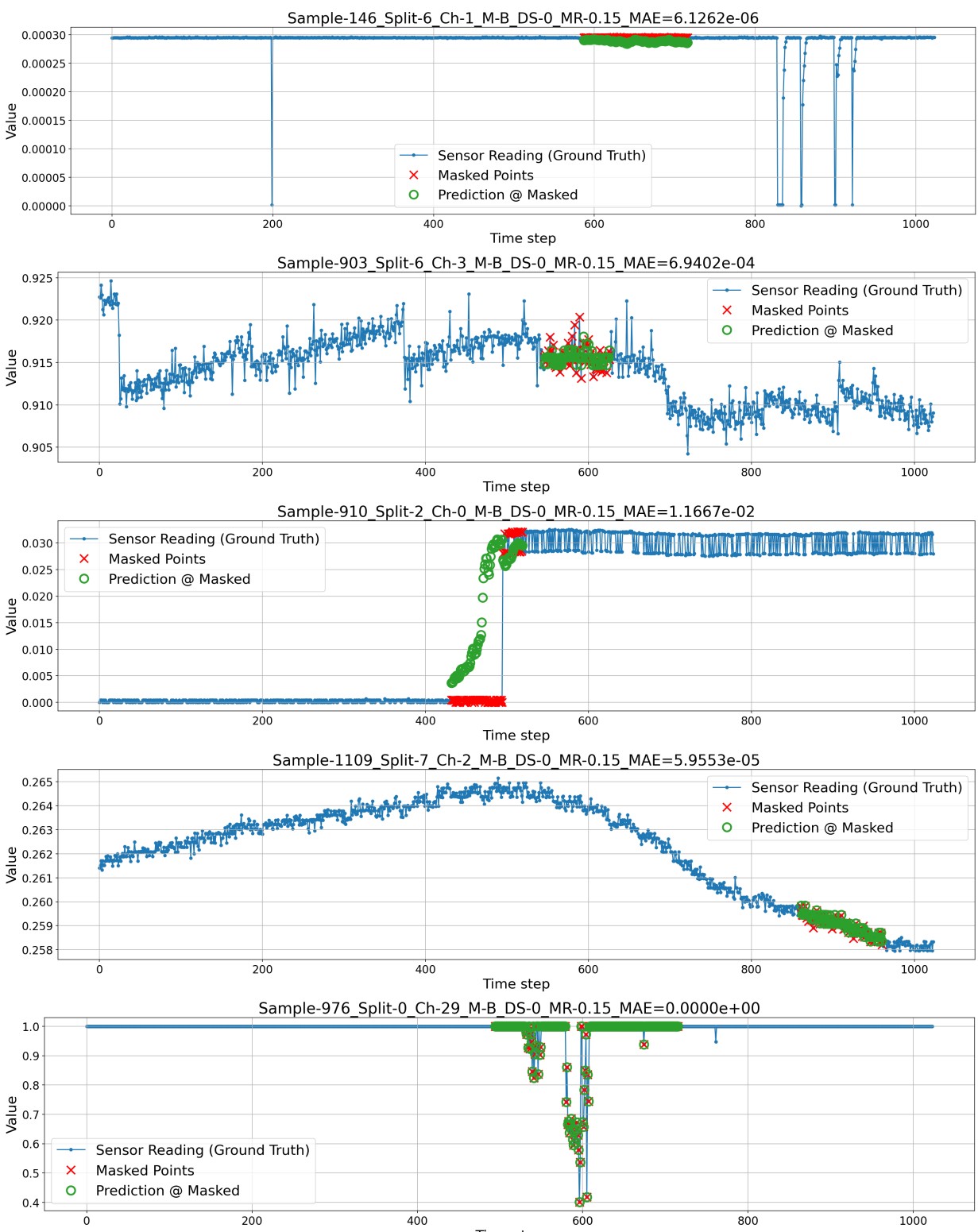

*Figure 5.* Representative zero-shot *block-wise imputation* examples from pretrained *SensOFormer-B* on held-out *LANSCE 2019–2024* test windows with a *15% mask ratio*. Blue curves show the normalized ground-truth sensor values, red markers indicate masked entries, and green circles show the model predictions at the masked locations. Each panel corresponds to a different test window and channel. The subplot title reports the sample index, split, channel, model size (M), dataset (DS), mask ratio (MR), and mean absolute error (MAE) computed on the masked entries.

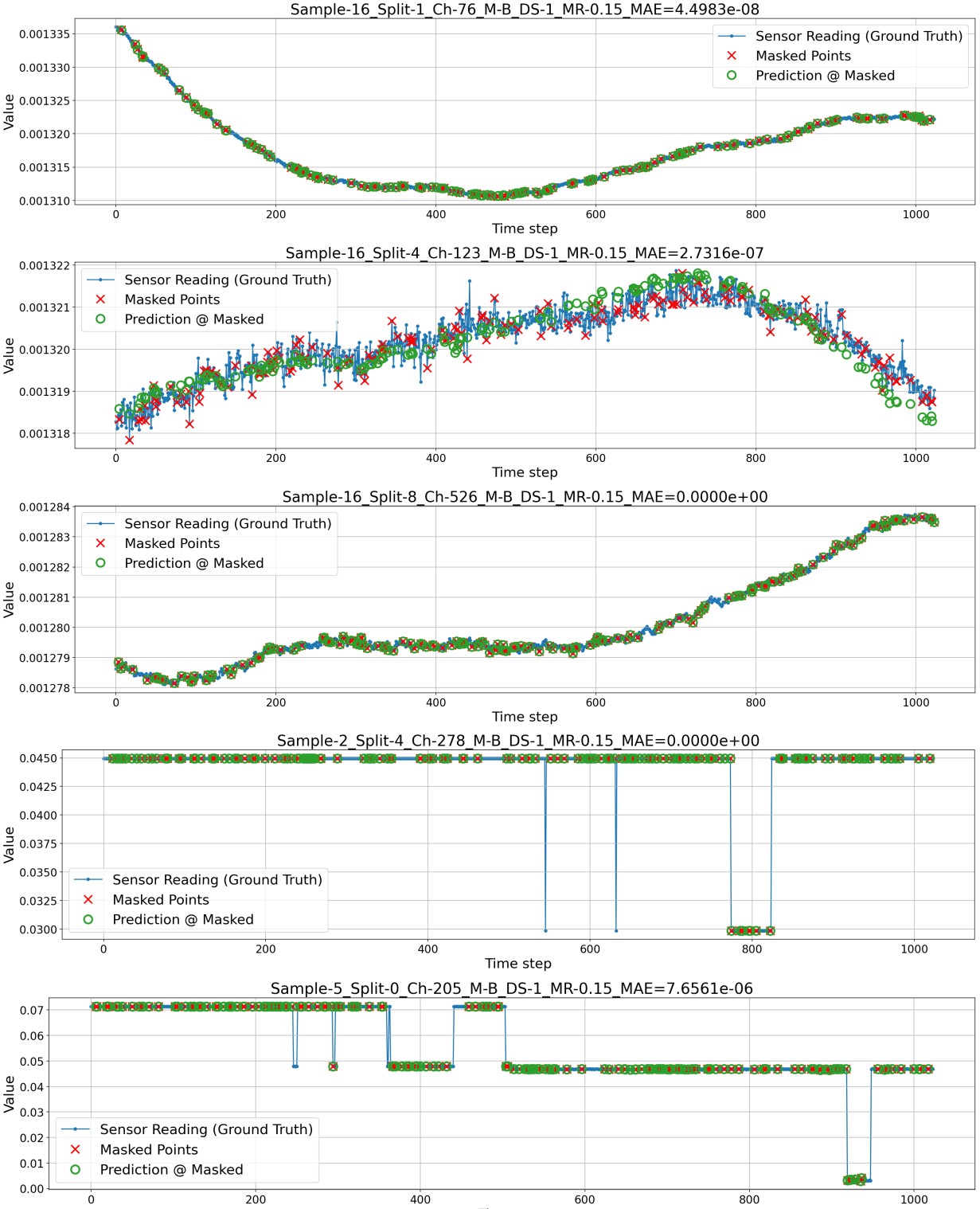

*Figure 6.* Representative zero-shot *point-wise imputation* examples from pretrained *SensOFormer-B* on held-out *LANSCE 2025* test windows with a *15% mask ratio*. Blue curves show the normalized ground-truth sensor values, red markers indicate masked entries, and green circles show the model predictions at the masked locations. Each panel corresponds to a different test window and channel. The subplot title reports the sample index, split, channel, model size (M), dataset (DS), mask ratio (MR), and mean absolute error (MAE) computed on the masked entries.

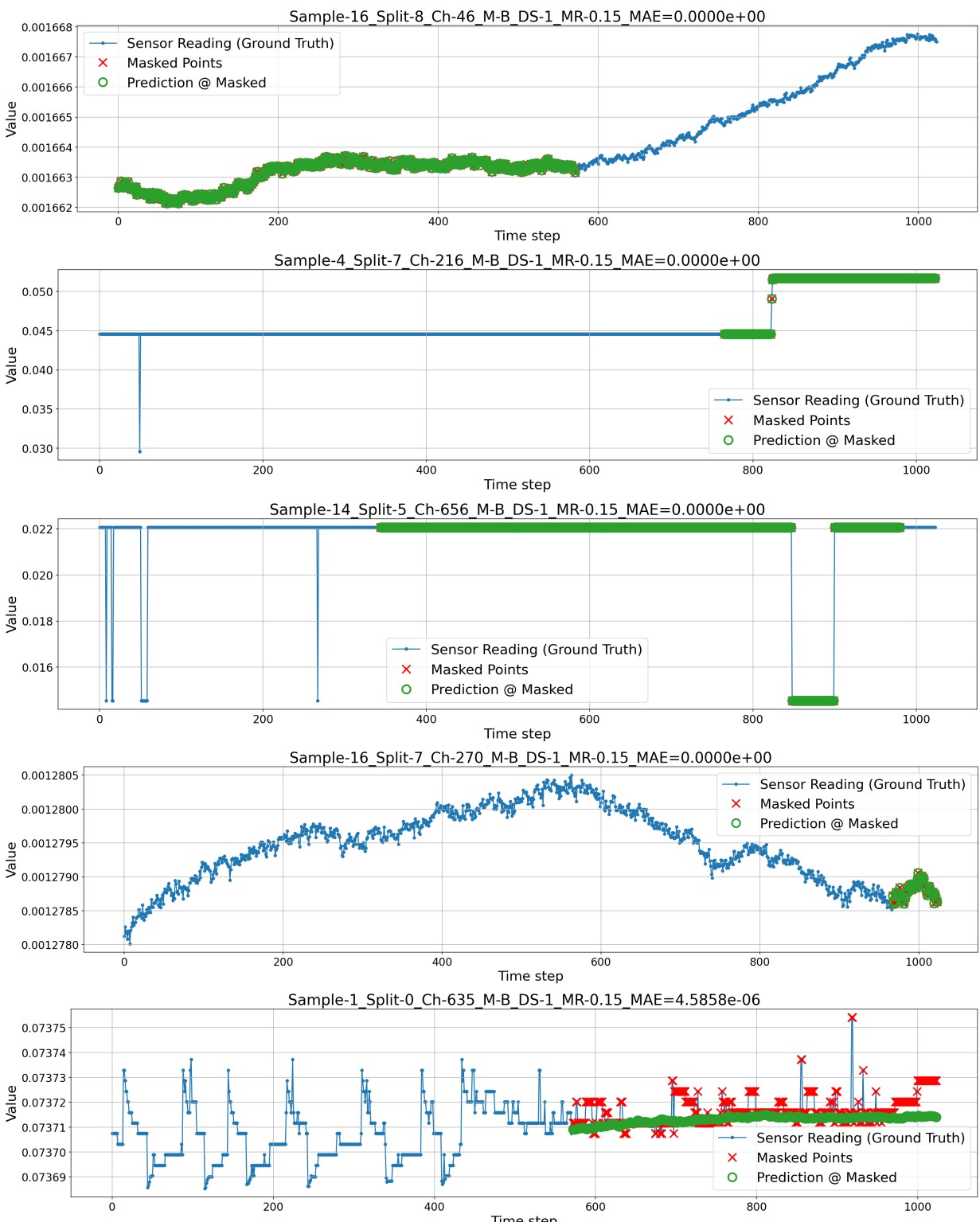

*Figure 7.* Representative zero-shot *block-wise imputation* examples from pretrained *SensOFormer-B* on held-out *LANSCE 2025* test windows with a *15% mask ratio*. Blue curves show the normalized ground-truth sensor values, red markers indicate masked entries, and green circles show the model predictions at the masked locations. Each panel corresponds to a different test window and channel. The subplot title reports the sample index, split, channel, model size (M), dataset (DS), mask ratio (MR), and mean absolute error (MAE) computed on the masked entries.

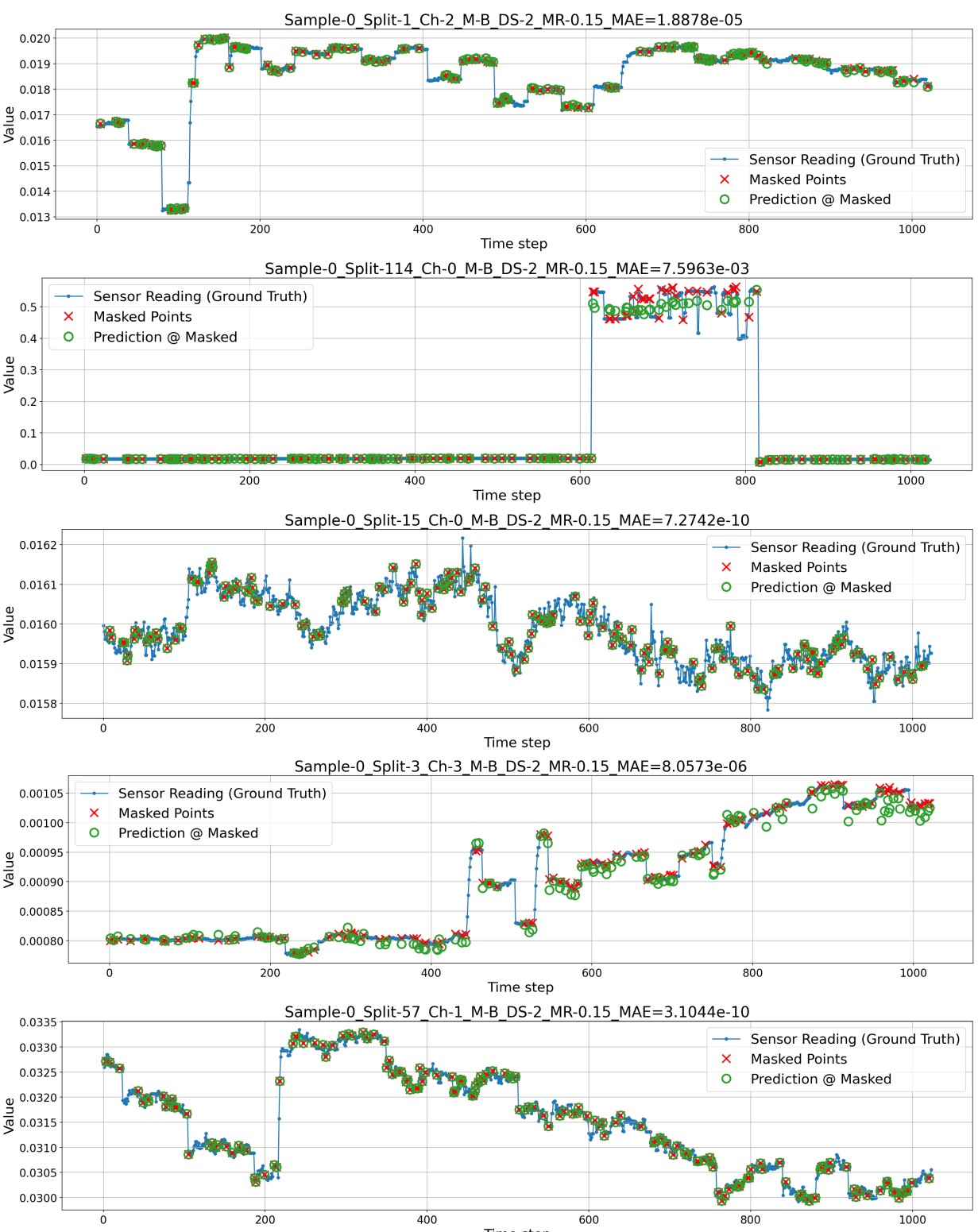

*Figure 8.* Representative zero-shot *point-wise imputation* examples from pretrained *SensOFormer-B* on held-out *LHC-BL* test windows with a *15% mask ratio*. Blue curves show the normalized ground-truth sensor values, red markers indicate masked entries, and green circles show the model predictions at the masked locations. Each panel corresponds to a different test window and channel. The subplot title reports the sample index, split, channel, model size (M), dataset (DS), mask ratio (MR), and mean absolute error (MAE) computed on the masked entries.

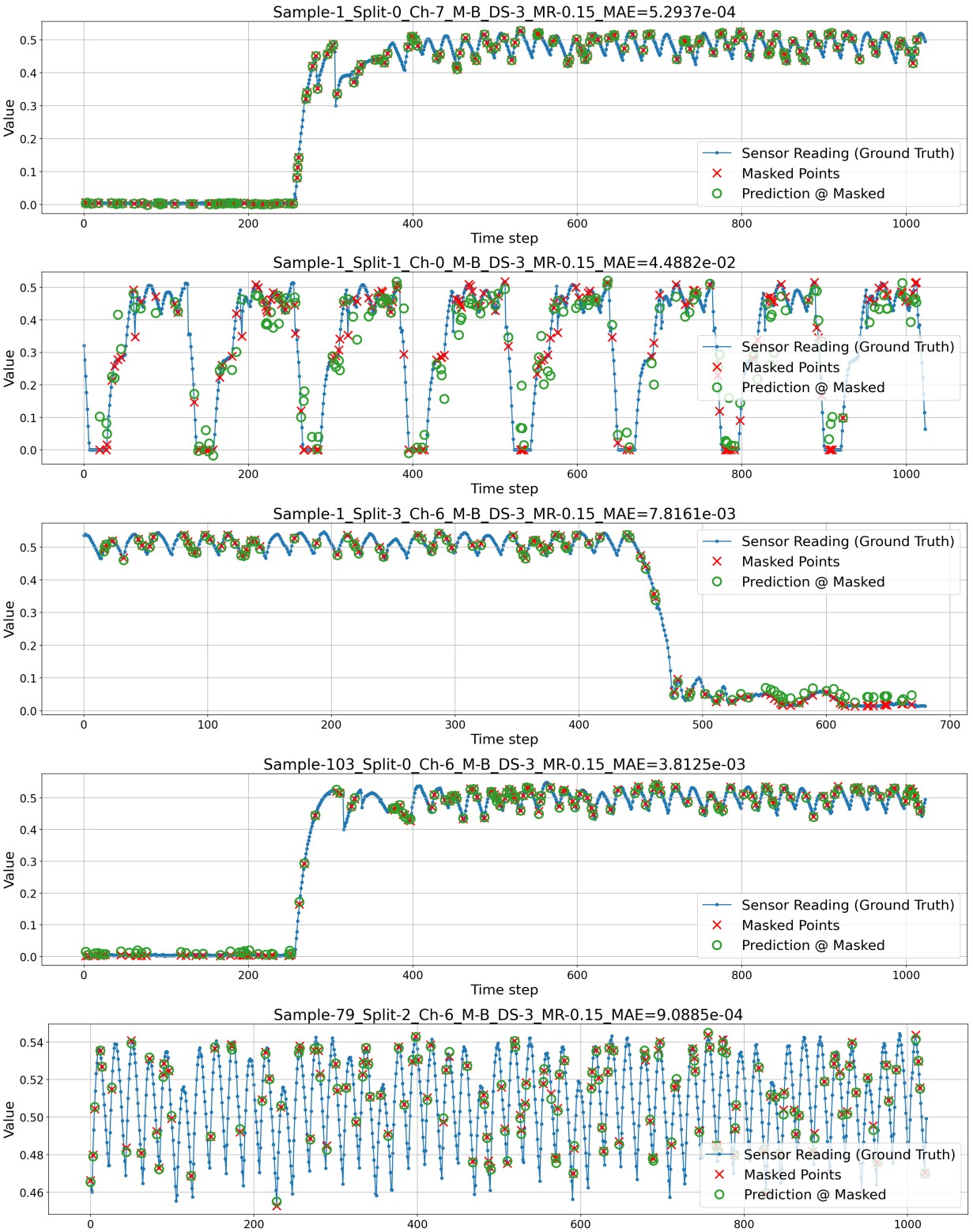

*Figure 9.* Representative zero-shot *point-wise imputation* examples from pretrained *SensOFormer-B* on held-out *HVCM-SNS* test windows with a *15% mask ratio*. Blue curves show the normalized ground-truth sensor values, red markers indicate masked entries, and green circles show the model predictions at the masked locations. Each panel corresponds to a different test window and channel. The subplot title reports the sample index, split, channel, model size (M), dataset (DS), mask ratio (MR), and mean absolute error (MAE) computed on the masked entries.

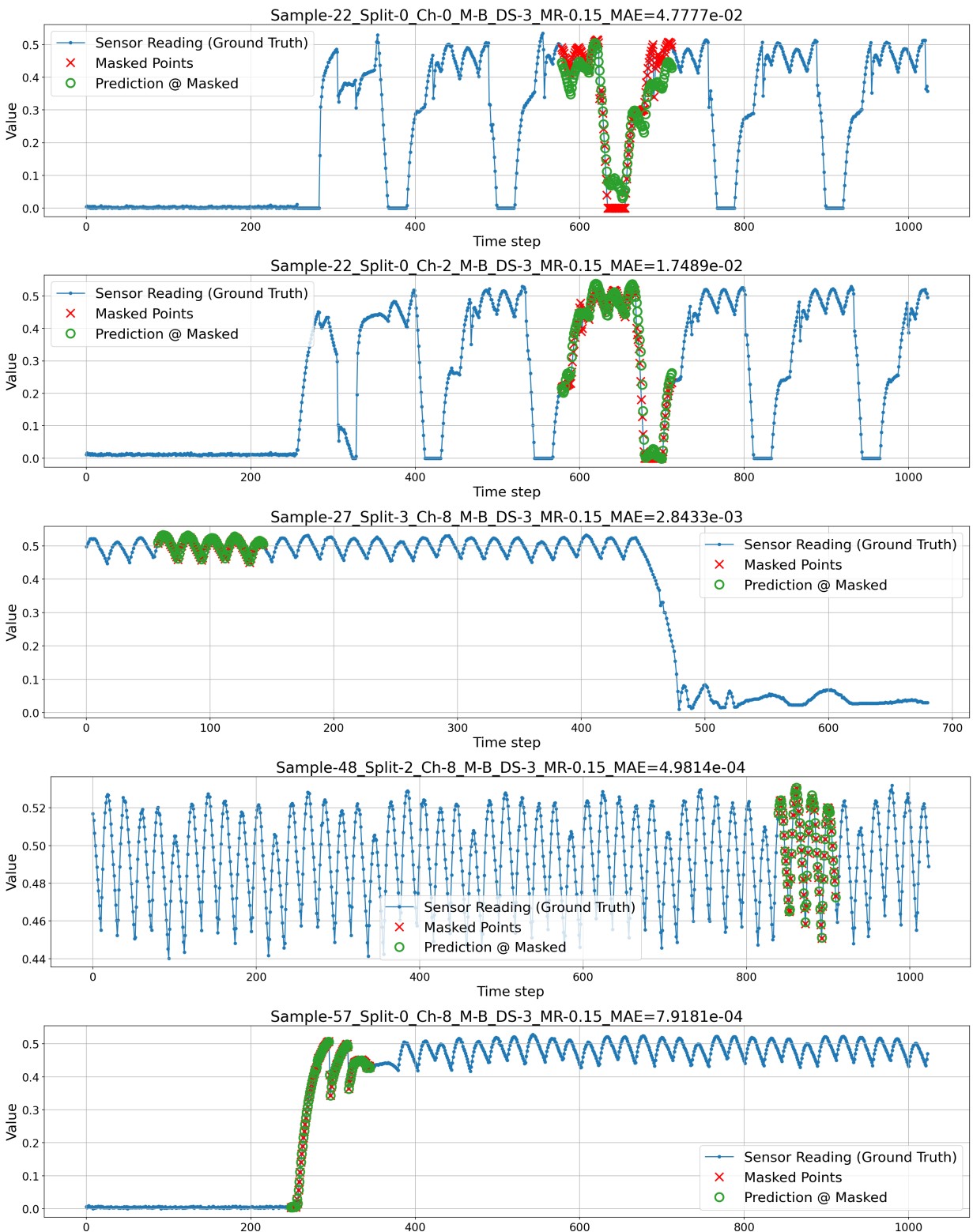

*Figure 10.* Representative zero-shot *block-wise imputation* examples from pretrained *SensOFormer-B* on held-out *HVCM-SNS* test windows with a *15% mask ratio*. Blue curves show the normalized ground-truth sensor values, red markers indicate masked entries, and green circles show the model predictions at the masked locations. Each panel corresponds to a different test window and channel. The subplot title reports the sample index, split, channel, model size (M), dataset (DS), mask ratio (MR), and mean absolute error (MAE) computed on the masked entries.

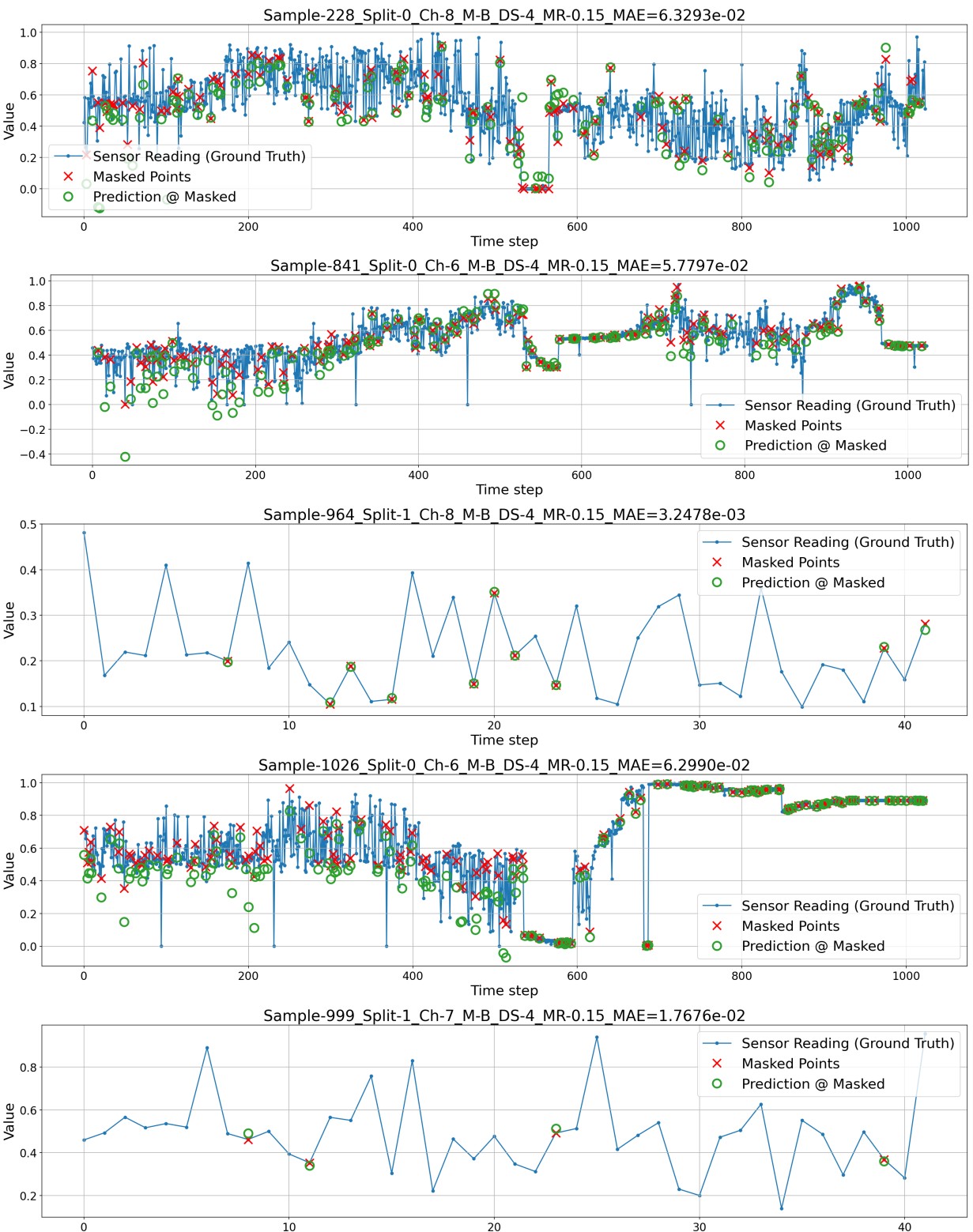

*Figure 11.* Representative zero-shot *point-wise imputation* examples from pretrained *SensOFormer-B* on held-out *RFS-LCLS* test windows with a *15% mask ratio*. Blue curves show the normalized ground-truth sensor values, red markers indicate masked entries, and green circles show the model predictions at the masked locations. Each panel corresponds to a different test window and channel. The subplot title reports the sample index, split, channel, model size (M), dataset (DS), mask ratio (MR), and mean absolute error (MAE) computed on the masked entries.

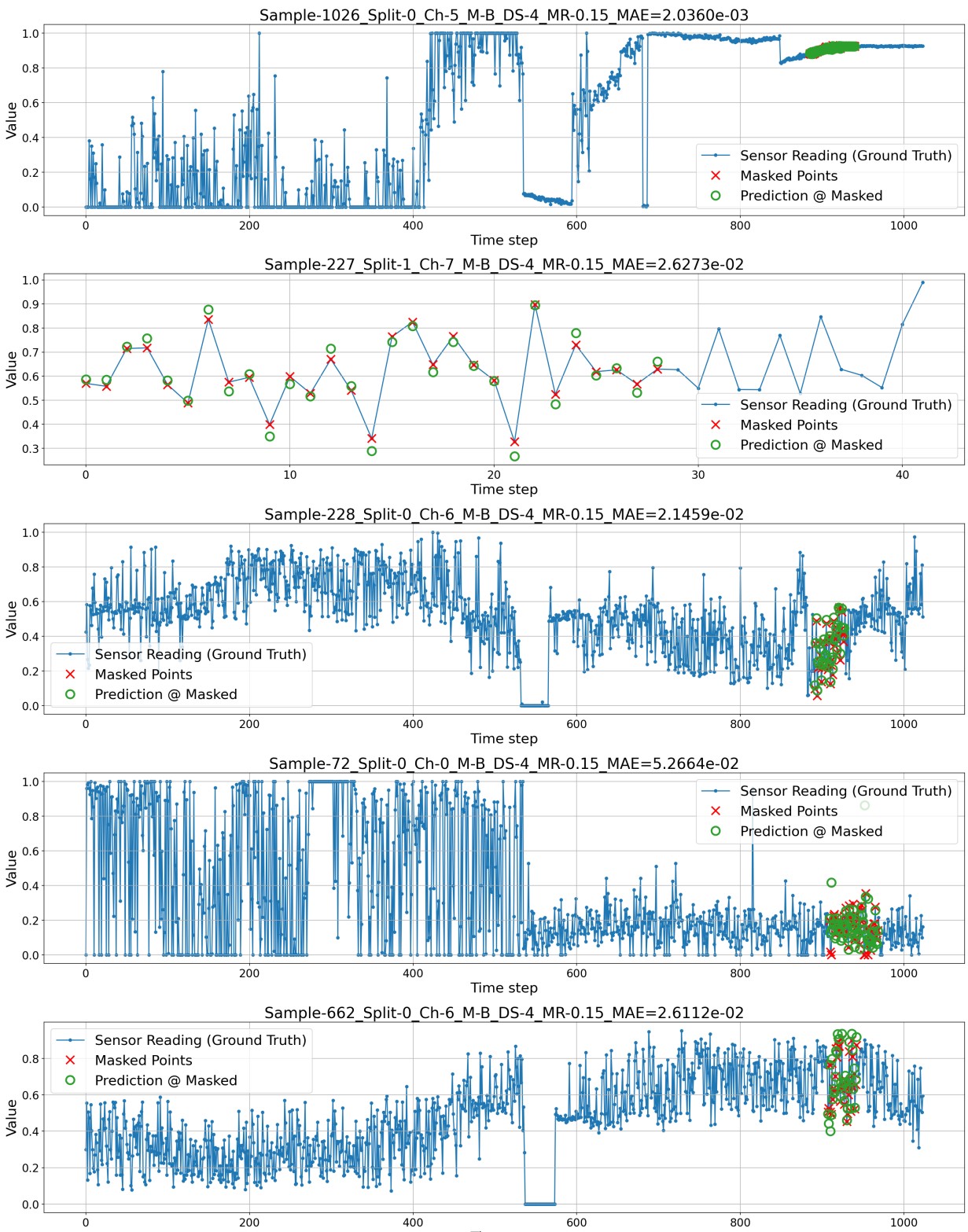

*Figure 12.* Representative zero-shot *block-wise imputation* examples from pretrained *SensOFormer-B* on held-out *RFS-LCLS* test windows with a *15% mask ratio*. Blue curves show the normalized ground-truth sensor values, red markers indicate masked entries, and green circles show the model predictions at the masked locations. Each panel corresponds to a different test window and channel. The subplot title reports the sample index, split, channel, model size (M), dataset (DS), mask ratio (MR), and mean absolute error (MAE) computed on the masked entries.

