# OpenReview forum: "A Foundation Model Approach to Particle Accelerator Operational Data"
_ICML.cc/2026/Workshop/FMSD — FMSD @ ICML 2026 Poster_

### Official Review · Reviewer_ar8S · 2026-05-20
**A Foundation Model Approach to Particle Accelerator Operational Data**

**Rating:** 9
**Confidence:** 4

**Review:**

# Summary
The authors introduce a new transformer model known as SensOFormer, pretrained on a number of particle accelerator tasks. The machinery behind particle accelerators is known to be very noisy, and could benefit from a model that supports many downstream tasks. SensOFormer was shown to work on a number of tasks such as missing data reconstruction, anomaly detection, fault identification and cavity identification.

# Strengths
+ The authors provide a pretrained foundation model capable of many downstream tasks.
+ Besides the Voting Ensemble, SOF outperforms every other architecture/method on early fault tests.
+ SOF outperforms all other methods on downstream tasks like fault and cavity detection.
+ Authors provide a scaling study showing loss correlates with data volume

# Areas for Improvement
- An ablation study on the various features and/or datasets would be a nice addition to the scaling study.
- Would be nice to see zero-shot generalization experiments from other datasets and/or accelerators.

# Detailed Comments
It would be nice to see a small diagram of the model in the main paper. I think that some of the tables or even figure 1 could be swapped out for this in the appendix.

# Justification of Score
Overall this is a really well put together paper and fits the venue well. The fact that the authors are able to provide a FM that outperforms previous SOTA in downstream tasks alone puts this in the top 15% of papers.

---

### Official Review · Reviewer_C1dk · 2026-05-21
**A telemetry foundation model is very beneficial to the community, but additional benchmarking should be included**

**Rating:** 7
**Confidence:** 4

**Review:**

The manuscript presents a foundation model for particle accelerator telemetry. Incorporating instrumental and detector data is highly valuable for scientific applications. The authors compare their approach against several traditional benchmarks and add a lot of detailed studies in the appendix. However, the evaluation does not include comparisons with established time-series foundation models (e.g., MOMENT), making it difficult to assess the relative performance of the proposed approach within the broader literature. The authors should also better discuss what the specific novelties of the sensoformer architecture are, rather than applying multi-head attention to physics data.

It is also unclear whether all datasets used in the experiments are derived from real accelerator operations or whether some are simulated. This distinction is important for evaluating the results, as simulated data sets are much easier.

In several places, the manuscript relies heavily on references to the appendix when details should instead appear in the main text. For example, in lines 156–159, different model variants (e.g., XTi, Ti) are introduced without sufficiently explaining their architectural differences beyond parameter counts. This should be adjusted. I also strongly recommend including a model architecture figure in the main paper. It would also be interesting to better study the latent space.

Given the importance of data-driven instrument models and the very interesting first results, I recommend the paper for acceptance.

---

### Official Review · Reviewer_dHgD · 2026-05-21
**Solid domain application of existing foundation model components, but limited architectural novelty and insufficient ablation support**

**Rating:** 5
**Confidence:** 4

**Review:**

SensOFormer is a self-supervised foundation model for particle accelerator operational telemetry. It combines a Perceiver-style encoder-decoder architecture with a masked denoising pretraining objective to learn sensor representations from heterogeneous, noisy accelerator data. The authors pretrain on data from five facilities (LANSCE, LHC, SNS, LCLS, CEBAF) and demonstrate transfer to four downstream tasks: missing-data imputation, anomaly detection, cavity identification, and fault identification. A scaling study shows reconstruction loss decreases with both model size and pretraining data volume.

Strengths
1. Well-motivated problem. Accelerator telemetry is genuinely challenging: heterogeneous sensors, variable sampling rates, noisy measurements, and scarce labeled fault data. The case for a shared foundation model backbone is well-argued and practically significant.
2. Multi-facility benchmark. Pretraining across five facilities with meaningfully different sensor modalities and temporal structures is a more demanding generalization test than most prior single-facility work in this area.
3. Label efficiency. SensOFormer surpasses prior deep learning baselines on fault identification using only 25% of labeled training data. This directly addresses a real operational constraint.
4. Anomaly detection. Matching the hierarchical voting ensemble on 20/21 early-fault tests while substantially outperforming individual classical and CNN baselines is a strong, practically meaningful result.


Areas for Improvement
1. Limited architectural novelty. The core components — Perceiver IO, standard transformer, masked autoencoder pretraining, and denoising objectives — are all well-established. The contribution is primarily domain application rather than architectural innovation, and the novelty framing should reflect this more honestly.
2. Architecture buried in the appendix. There is no dedicated methods section in the main paper. The model description is spread across Appendix B with the only architectural diagram on page 9. For a paper whose central contribution is a new model, this is a significant structural weakness.
3. No ablation study. The paper does not isolate the contribution of individual design choices. Does the Perceiver-style encoder outperform a simpler fixed-channel encoder? How much does the multi-type mask bank contribute versus a single masking strategy? Without these, it is unclear which aspects of the design drive the reported gains.
4. Weak imputation baselines. For the primary downstream task, comparisons are limited to SensOFormer size variants only. No established time-series imputation baselines such as SAITS or BRITS are included.

Detailed Comments

The novelty claim of being "the first foundation model for particle-accelerator telemetry" is a domain-priority claim, not an architectural one. The authors should frame contributions more explicitly as: domain adaptation of existing techniques, a new multi-facility benchmark, and empirical transfer evidence.
All supervised downstream experiments use only the smallest model variant (XTi, 2.38M parameters), despite imputation results showing clear gains from larger variants. This choice needs explicit justification.
The LANSCE-2025 distribution shift results show substantially higher errors but receive no meaningful analysis. For practitioners this is an important failure mode that deserves more attention.
The data-scarce fault identification curve would benefit from confidence intervals across multiple runs, particularly at low data fractions where variance is likely high.


Justification of Score
SensOFormer is a competent and practically motivated paper with a strong multi-facility benchmark and a compelling label efficiency result. However, architectural novelty is limited, the core technical content is entirely in the appendix, and the experimental validation has meaningful gaps — no external imputation baselines and no ablations. The domain application is relevant to the workshop theme, but the paper needs stronger empirical grounding to be fully convincing.
Rating: 5 — Marginally below acceptance threshold
Confidence: 4 — Confident but not absolutely certain